# Cavitation Characterization of Size-Isolated Microbubbles in a Vessel Phantom Using Focused Ultrasound

**DOI:** 10.3390/pharmaceutics14091925

**Published:** 2022-09-12

**Authors:** Payton Martinez, Nick Bottenus, Mark Borden

**Affiliations:** 1Biomedical Engineering Program, University of Colorado, Boulder, CO 80309, USA; payton.martinez-1@colorado.edu (P.M.); nick.bottenus@colorado.edu (N.B.); 2IQ Biology Program, University of Colorado, Boulder, CO 80309, USA; 3Mechanical Engineering Department, University of Colorado, Boulder, CO 80309, USA

**Keywords:** passive cavitation detection, size-isolated microbubbles, focused ultrasound, stable cavitation, inertial cavitation

## Abstract

Pharmaceutical delivery can be noninvasively targeted on-demand by microbubble (MB) assisted focused ultrasound (FUS). Passive cavitation detection (PCD) has become a useful method to obtain real-time feedback on MB activity due to a FUS pulse. Previous work has demonstrated the acoustic PCD response of MBs at a variety of acoustic parameters, but few have explored variations in microbubble parameters. The goal of this study was to determine the acoustic response of different MB size populations and concentrations. Four MB size distributions were prepared (2, 3, 5 µm diameter and polydisperse) and pulled through a 2% agar wall-less vessel phantom. FUS was applied by a 1.515 MHz geometrically focused transducer for 1 ms pulses at 1 Hz PRF and seven distinct mechanical indices (MI) ranging from 0.01 to 1.0 (0.0123 to 1.23 MPa PNP). We found that the onset of harmonic (HCD) and broadband cavitation dose (BCD) depends on the mechanical index, MB size and MB concentration. When matched for MI, the HCD and BCD rise, plateau, and decline as microbubble concentration is increased. Importantly, when microbubble size and concentration are combined into gas volume fraction, all four microbubble size distributions align to similar onset and peak; these results may help guide the planning and control of MB + FUS therapeutic procedures.

## 1. Introduction

Microbubble ultrasound contrast agents (MBs) typically comprise a phospholipid shell and a low-solubility weight gas (e.g., perfluorocarbons) [1] and range from 0.1 to 10 um in diameter [2,3]. Several compositions are clinically approved for ultrasound contrast imaging due to their echogenicity [3]. The stabilizing shell can be modified to provide ligand targeting or drug attachment [4]; this versatility has allowed the emergence of microbubbles as an agent for more precise imaging applications, including super-resolution and molecular imaging [5,6,7].

A unique property of microbubbles is their ability to cavitate (oscillate) strongly under ultrasound. The MB oscillations in response to ultrasound induce localized forces that can provide a means of cell membrane permeabilization and drug extravasation [8,9,10]. The combination of these effects with the noninvasive targeting precision of focused ultrasound (FUS) has become an area of interest for novel pharmaceutical therapeutics; it has been shown that FUS and MBs can disrupt the blood–brain barrier (BBB) at a lower mechanical index (MI) than for FUS alone [11,12,13]. At the lower MI, there is minimal damage while creating a pathway for macromolecules to enter the parenchyma [8,14]. To create safer and more effective therapies, efforts are underway to better understand and control the mechanisms behind MB cavitation.

Many acoustic parameters have been investigated with respect to the resulting cavitation. For example, the pressure threshold of blood–brain barrier disruption (BBBD) was shown to decrease with increasing pulse length, where an increasing number of cycles increased the extent of BBBD [11,15]. However, the pulse repetition frequency has not been shown to affect the threshold [11,12]. The mechanical index is one of the most important factors for achieving BBBD [11,12,13]. While these studies have varied the ultrasound parameters, few have discussed the microbubble parameters.

Microbubble size has been of interest to the effect on therapy, including BBBD; it was previously shown that dye-delivery by BBBD depends on microbubble diameter [16,17]. Larger microbubbles had a higher BBBD when concentrations were matched by the number of bubbles; these studies demonstrated that larger microbubbles had a lower threshold to observe significant BBBD [16,17]. Wang et al. Furthermore showed that microbubble composition (e.g., choice of gas core, shell) affects the intensity of BBBD, although differences were not as significant as size [18].

Clinically approved microbubbles come in a variety of concentrations and size distributions that can even vary vial-to-vial for the same product [19]. Studies have shown that this variety may result in different acoustic responses in both imaging [20,21,22,23] and therapeutics, which can result in different biological effects [16,24,25]. MB size distribution and concentration can be combined into a single parameter, called the gas volume fraction (GVF, µL/mL), or microbubble volume dose (MVD) when prescribed to the subject’s body weight (µL/kg) [26]. Interestingly, effects of different MB size distributions at matched MVD were shown to be similar for both pharmacokinetics [25,27] and BBBD drug extravasation [26]; this begs the question as to whether cavitation activity of different MB size distributions and concentrations also collapses to a master curve when plotted against MVD. Such a result would help to bridge the knowledge gap between MB pharmacokinetics and MB + FUS bioeffects.

A passive cavitation detector (PCD) allows real-time analysis of the acoustic emissions of the microbubbles [28,29,30,31,32]. Previous reports have shown two defining forms of MB cavitation, “stable” and “inertial” cavitation [33]. Stable cavitation occurs at lower acoustic pressures and is defined by relatively small oscillations and minimal effect on the resting size of the microbubble [33,34,35]. Stable cavitation has been identified by harmonic emissions during ultrasonic excitation [4,36]. However, harmonic signals can be produced by tissue or coupling media and therefore are not strictly indicative of microbubbles [37]. On the contrary, sub/ultra-harmonic emissions only arise from microbubbles and are, therefore better indicators of bubble activity [38]. As the acoustic driving pressure increases, the MBs can move into the inertial cavitation regime. Inertial cavitation emissions have been identified by a broadband PCD response [28,30,36], where they collapse violently to produce a broadband echo. At this point, jetting, shockwaves and other inertial effects can occur [33,35,39].

However, microbubbles observed under high-speed video microscopy have been shown to display relatively large oscillations and may change the size by rectified diffusion, coalescence, fragmentation or dissolution at the same acoustic forcing in which they would be characterized as undergoing “stable” cavitation with PCD data [40,41,42]. In fact, the MBs may be dynamic (changing size and number), rather than stable, while they are emitting sub/ultra-harmonic echoes. MBs can also move into intermittent inertial cavitation and still produce sub and ultra-harmonic echoes [43]. We, therefore, use the terms “harmonic” or “broadband” cavitation, rather than “stable” or “inertial” cavitation, to describe the behavior.

The purpose of this study was to investigate the effects of microbubble size and concentration on the harmonic and broadband cavitation response in a vessel phantom. To achieve this, microbubbles were pulled through a wall-less agar vessel phantom while FUS was applied. The resulting acoustic response was recorded with a PCD. Four size distributions were analyzed at mechanical indices ranging from 0.01 to 1.0. Each size distribution was also examined at a range of concentrations from 1.5 × 10^4^ to 1.5 × 10^8^ MBs/mL. The harmonic cavitation dose (HCD) and broadband cavitation dose (BCD) were calculated and compared at different MB diameters, concentrations and GVFs.

## 2. Materials and Methods

### 2.1. Materials

All solutions were made using filtered, sterile, deionized water (Direct-Q 3 Millipore, Billerica, MA, USA). All glassware was cleaned with 70 vol% isopropyl alcohol (Supelco, Burlington, MA, USA). 1,2-distearoyl-sn-glycero-3-phosphocholine (DSPC) was purchased from Avanti Polar Lipids (Alabaster, AL, USA). Perfluorobutane gas (n-C_4_F_10_, PFB) was purchased from FluroMed (Round Rock, TX, USA). Polyoxyethylene-40 stearate (PEG40S) and chloroform were purchased from Sigma-Aldrich (St. Louis, MO, USA). Sterile saline solution and phosphate-buffered saline solution (PBS) were purchased from Fisher Scientific (Pittsburg, PA, USA). The purity of all the reagents was ≥99%, and they were used as received without further purification.

### 2.2. Experimental Set-up

To target focused ultrasound, we designed and built a custom wall-less vessel phantom (Figure 1). The RK-50 system (Stereotactic-guided Focused Ultrasound, FUS Instruments, Toronto, ON, Canada) was used for our experiments. The RK-50 system has a single element, 1.515 MHz, focused transducer with a 20-mm diameter. To receive returning signals from the microbubbles, a built-in passive cavitation detector (PCD) positionedcoaxially within the focused transducer was used. The PCD had a wide bandwidth and center frequency of 0.7575 MHz, with an 8 mm diameter. PCD voltage vs. time data were collected by the RK-50 software. The phantom was placed under the transducer. The phantom shell was filled with filtered and degassed water. On either end of the vessel, female luers were used to attach 0.9 mm outer diameter tubing. One end of the tubing was placed in a beaker of microbubbles on a stir-plate with medium stirring (150 RPM). The other end of the tubing was attached to a 12 mL syringe (Monoject, Coviden, Mansfield, MA, USA) on a withdrawing syringe pump (Model 1000, New Era Pump Systems, Farmingdale, NY, USA). Microbubbles were diluted down to the desired concentration just before the flow was started.

### 2.3. Fabrication of Agar Phantom

The agar phantom was created within an acrylic box with dimensions shown in Appendix A. Barbed female luers were added to four evenly spaced points along the length of the box with barbed pieces facing inwards. A sorbothane vibration isolating rubber (Isolate It! Burlington, NC, USA) layer was added to the bottom of the box as an acoustic absorber. Prior to preparing the agar, polyethylene tubing (Warner Instruments, Holliston, MA, USA) with an outer diameter of 0.9 mm was pulled through both ends of the luers. The 0.9 mm diameter was chosen to be small enough to be clearly within the focal zone of the FUS and PCD. Agar (2%) was prepared using 11 g of Agar powder (Millipore, Burlington, MA, USA) mixed with 550 mL of filtered and distilled water at room temperature for 20 min or until the powder had fully dissolved. The solution was then placed in an 1100-Watt microwave and heated for 30 s, mixed, then reheated until it was transparent. The solution was then placed in a vacuum chamber to degas. The pressure was slowly reduced by 80 kPa over a five-minute period. The solution was then removed and slowly poured into the acrylic box. Polyethylene tubing was adjusted as needed to confirm a perpendicular angle to the side of the phantom. All, if any, air bubbles were removed by popping or scooping them out. The phantom was allowed to cool for one and a half hours at room temperature.

### 2.4. Microbubble Preparation

Lipid-coated MBs with a PFB gas core was prepared by sonication, as described previously [44]. Briefly, under sterile conditions, a dried lipid film comprising DSPC:PEG40S (90:10) was hydrated with filtered and sterile PBS (1X) to a final lipid/emulsifier concentration of 2 mg/mL at 65 °C for 40 min. Once fully hydrated, the solution was allowed to cool to room temperature. The lipid solution was sonicated with a 20 kHz probe (model 250A, Branson Ultrasonics; Danbury, CT, USA) at low power (3/10; 3 W) for four minutes. After cooling to room temperature, PFB was delivered to the surface of the lipid suspension for 10 s. Then, the solution was sonicated at high power (10/10; 33 W) for 30 s to produce polydisperse MBs. Polydisperse MBs were then collected into 30 mL sterile syringes and isolated by differential centrifugation into three diameters: 2, 3 and 5 µm diameter. The size isolation process, including time and speeds of centrifugation, can be found in Appendix A. MB concentration and number- and volume-weighted size distributions were measured using a Multisizer 3 Coulter Counter (Beckman Coulter). MB concentration (
c
, MBs/µL) versus MB volume (
v
, µL/MB) was plotted, and MB gas volume fraction (
ΦMB
) was estimated as follows:
(1)
$$\Phi MB=\overset{n}{\underset{i=1}{\sum }}c_{i}\times v_{i}$$

where 
i
 is the index of the sizing bin, and there were 300 logarithmically spaced bins ranging from 0.7 to 18 µm in diameter. Three independent size-isolated *MB* preparations were evaluated after synthesis and three hours prior to sonication to confirm the size distributions and concentration. Finally, after isolation, *MB* cakes were stored in the refrigerator at 4 °C for subsequent use.

### 2.5. Focused Ultrasound Stimulation of Microbubbles

A sinusoidal burst signal with a center frequency of 1.515 MHz was produced using the RK-50 system. The signal had a pulse repetition frequency (PRF) of 1 Hz, pulse duration (PD) of 1 ms, the peak negative pressures (*PNP*) varied from 12.3 kPa to 1.23 MPa. Mechanical Index (*MI*) is a unifying ultrasound parameter that combines the ultrasound center frequency (*f* in units of MHz) and peak negative pressure (*PNP* in units of MPa). The relationship is:
(2)
$$MI=\frac{PNP}{\sqrt{f}}$$


Previous studies have shown that *MI* correlates well with biological effects including BBBD over a range of peak negative pressures and fundamental frequencies [12]. Therefore, the *MI*s used during sonication were between 0.01 and 1.0. Microbubbles were kept at 4 °C just prior to dilution into room temperature water and subsequent PCD measurement. MB stability during stirring was confirmed by measuring the concentration and mean diameter periodically after 15 min of stirring (Appendix A). The syringe pump withdrew at 34 mL/h to allow new microbubbles in the focal area of ultrasound between each pulse. The flowrate was slow enough to avoid a large pressure gradient (~150 Pa), yet high enough to avoid significant bubble flotation on the top of the tube due to buoyancy.

### 2.6. PCD Data Collection and Analysis

To analyze the acoustic cavitation of the microbubbles, returning pressure was converted into voltage via a PCD device built into the FUS transducer. The resulting voltage over time signals were stored on the RK-50 desktop. Data were collected at a 37 MHz sampling frequency. The collection started just prior to initial sonication. All data were analyzed using MATLAB (MathWorks, Natick, MA, USA). Voltage data were first cropped to remove any signal prior to the first expected return signal (pre-signal sensitivity plus travel time of the first pulse). Tukey windowing was applied to the cropped signal to prevent spectral leakage on the fast Fourier transform (FFT). The FFT was then taken for every pulse (26). The FFT magnitudes were averaged together within each treatment.

### 2.7. Harmonic Cavitation Dose

Sub-harmonic and ultra-harmonic (F × n/2; *n* = 1, 3, 5…) frequencies are associated with the harmonic (non-broadband) cavitation of microbubbles. Once voltage data were converted to the frequency domain, the power spectrum was used to analyze these frequencies. A lowpass filter was applied at the end of the fourth harmonic (6.06 MHz). The sub-harmonic and the first three ultra-harmonic (e.g., 0.7575, 2.2725, 3.7875, 5.3025 MHz) components were extracted using a minimum order bandpass filter, with a bandwidth of 0.1 MHz, at each frequency. Fast Fourier transform was used to obtain the frequency spectrum:
(3)
$$X\left(k\right)=\overset{N-1}{\underset{i=0}{\sum }}x\left(i\right)e^{-j\frac{2\pi }{N}ik}\left(k=0,1,\ldots ,N-1\right)$$

where *k* is the frequency point in the frequency spectrum. To determine the intensity of the harmonic frequency components, the power spectrum of the signal was then calculated by:
(4)
$$P\left(k\right)=X{\left(k\right)}^{2}\left(k=0,1,\ldots ,N-1\right)$$


The resulting power spectrums were summed together and multiplied by the total sampling time:
(5)
$$HCDm=Ts\;\overset{a}{\underset{1}{\sum }}\overset{b}{\underset{1}{\sum }}P\left(k\right)\;$$

where *a* and *b* are the number of sub/ultra-harmonic components and the number of frequency points in every sub/ultra-harmonic component, respectively. Note that the four sub/ultra-harmonic components ranged from 0.7075 MHz to 5.3525 MHz, (*a* = 1, 2, ⋯, 4), and the total number of frequency points in every component can be calculated by dividing the bandwidth of analysis (0.1 MHz) by the frequency resolution (1587 Hz), giving 63 points per component. In Equation (5), the coefficient was the sampling time (
Ts
) for harmonic cavitation signals. The sampling time used was 850 × 10^−6^ s. The real harmonic cavitation dose (*HCD*) from the microbubbles can be calculated by subtracting the measurement from the control degassed saline solution (*HCDc*, Appendix A):
(6)
HCD=HCDm−HCDc


### 2.8. Broadband Cavitation Dose

The broadband signal is associated with the inertial cavitation of microbubbles. Once voltage data were converted to the frequency domain, the power spectrum was used to analyze these frequencies. A lowpass filter was applied at the end of the fourth harmonic (6.06 MHz). The subharmonic, the first three ultra-harmonics and the first four harmonics (e.g., 0.7575, 1.515, 2.2725, 3.03... etc. MHz) components were removed using a stopband filter, with a bandwidth of 0.1 MHz, at each frequency. Fast Fourier transform was used to obtain the frequency spectrum:
(7)
$$X\left(k\right)=\overset{N-1}{\underset{i=0}{\sum }}x\left(i\right)e^{-j\frac{2\pi }{N}ik}\left(k=0,1,\ldots ,N-1\right)$$

where again *k* is the frequency point in the frequency spectrum. To determine the intensity of the broadband frequency components, the power spectrum of the signal was then calculated by:
(8)
$$P\left(k\right)=X{\left(k\right)}^{2}\left(k=0,1,\ldots ,N-1\right)$$


The resulting power spectrums were summed together and multiplied by the total sampling time:
(9)
$$BCDm=Ts\;\overset{a}{\underset{1}{\sum }}\overset{b}{\underset{1}{\sum }}P\left(k\right)$$

where *a* and *b* are the number of broadband components and the number of frequency points in every broadband component, respectively. Noted that seven broadband components ranged from 0.8075 MHz to 6.01 MHz, (*a* = 1, 2, ⋯, 7), and the total number (414) of frequency points in every broadband component can be calculated by dividing the bandwidth of analysis (0.6575 MHz) by the frequency resolution (1587 Hz). In Equation (9), the coefficient was the sampling time (
Ts
) for broadband cavitation signals. The sampling time used was 850 × 10^−6^ s. After broadband cavitation dose control (*BCDc*) from the control degassed saline solution was obtained, the real broadband cavitation dose (*BCD*) from the microbubbles can be calculated by:
(10)
BCD=BCDm−BCDc


### 2.9. Statistical Analysis

For each treatment a total of 26 bursts were completed. An FFT was taken for each burst individually and the magnitudes were averaged together. A minimum of four replicate treatments were done for each concentration, mechanical index, and microbubble size distribution using a new batch of microbubbles and agar. The saline control solution was also completed after changing microbubble size distribution, new agar, or moving to a new channel. The control values were ensured to be similar in magnitude for each change (Appendix A). The standard deviation and mean were calculated for the respective groups. An ANOVA was completed at all plots where *HCD* or *BCD* was expressed. Tukey’s multiple comparisons test was used to determine the significance between points. All adjusted *p*-values obtained are shown in Appendix A.

## 3. Results and Discussion

### 3.1. MB Characterization

Following size isolation of microbubble populations, each size was visually inspected using microscope brightfield images shown in Figure 2A–D. Coulter Multisizer sizing confirmed an initially broad polydisperse population with uniform sizes following centrifugal isolations. Figure 2E illustrates the number weighted percent of the populations giving a mean diameter of 2.0, 2.9, 4.4 and 1.9 µm for 2, 3, 5 µm and polydisperse respectively. Volume weighted percent had similar narrow peaks with mean diameters of 2.4, 3.5, 5.5 and 8.8 µm (Figure 2F). Figure 2G shows the plot used to determine the gas volume fraction (
ΦMB
) of each microbubble (MB) population. The mean 
ΦMB
 per MB for 10^10^ MBs/mL was shown to be 5.28, 14.8, 54.7, and 18.7 µL/mL for 2, 3, 5 µm and polydisperse respectively (Table 1). Statistical analysis was done for each plot, showing significant differences between all size isolated populations (*p* < 0.05).

### 3.2. Signal Analysis

The acoustic response of control PBS is shown in Figure 3. As expected, the saline showed very little acoustic feedback at all mechanical indices. Appendix A shows the difference in harmonic cavitation dose between the control PBS and the microbubbles. Figure 4 shows the voltage and frequency response of 5 µm diameter microbubbles (1.5 × 10^6^ MBs/mL; MI = 1.0) flowing (34 mL/h) through a 900 µm diameter wall-less vessel. After signal processing depicted in Figure 3 and Figure 4, two types of responses could be determined: harmonic cavitation dose (HCD) and broadband cavitation dose (BCD).

### 3.3. Harmonic Cavitation Dose vs. Mechanical Index

The subharmonic and ultraharmonic response of four different size distributions of microbubbles across a range of mechanical indices (MIs) from 0.01 to 1.0 is illustrated in Figure 5. For all diameters at a given concentration, the HCD increased only slightly up to a threshold MI, above which a larger increase was observed. The general trend was a linear increase for both regimes of HCD vs. MI, below and above the threshold MI. Our threshold MI for the onset of harmonic cavitation dose (HCD) matched previously determined MIs both in vivo [37] and in vitro [45]. The threshold MI appeared to increase in magnitude for decreasing bubble diameter (e.g., from 0.4 MI for 2 µm MBs to 0.1 MI for 5 µm MBs at a matched concentration of 1.5 × 10^5^ MBs/mL). Additionally, at a given MI, the HCD increased to a peak and then decreased with MB concentration, presumably due to attenuation of the US signal during both transmit and receive; this phenomenon can also be explained by bubble-bubble interactions during the sonication [46,47]. The peak HCD was at a concentration of ~10^7^ MBs/mL for 2 µm MBs and ~10^6^ MBs/mL for the other MB sizes.

### 3.4. Broadband Cavitation Dose vs. Mechanical Index

The broadband response of four different size distributions of microbubbles across a range of mechanical indices (MIs) from 0.01 to 1.0 is shown in Figure 6. There was a clear onset of BCD after 0.4 MI for all MB sizes, which matches previous literature [28,29,30]. After 0.4 MI, the BCD increased linearly, although at different slopes depending on the concentration. In some cases (e.g., for 1.5 × 10^8^ MBs/mL for 2 µm MBs), there was a slight reduction from the linear trend at the highest MI of 1.0, presumably due to MB destruction from pulse to pulse.

### 3.5. Harmonic Cavitation Dose vs. Concentration

Figure 7 illustrates the sub/ultraharmonic response as number concentration was increased. The HCD increased, peaked, and then declined as MB concentration increased. Comparing across size distributions, the larger microbubbles (5 µm) had a peak around 1.5 × 10^6^ MBs/mL whereas the smallest microbubble diameter (2 µm) had a peak around 1.5 × 10^7^ MBs/mL. Both polydisperse and 3 µm diameter distributions had a similar peak around 1.5 × 10^6^ MBs/mL. From a theoretical perspective, increasing HCD with MB concentration follows simply as an increase in the number of scatterers. The decreasing trend is likely due to attenuation (due to scattering and absorption) of the ultrasound signal during both transmit and receive. Previous work using numerical simulations has shown this decrease in sub/ultra-harmonic emissions as the number of microbubbles increases [48]. The simulations showed that bubble-bubble interactions intensified broadband signaling and therefore hid the harmonic emissions [46,48]. Similarly, using randomly distributed microbubbles, it was shown that sub and ultra-harmonic emissions plateau and decrease as the concentration of MBs is increased [47,49,50]; it is also reported that a peak acoustic emission occurs around 10^6^ MBs/mL, matching our results (Figure 7) [50]. Additionally, previous work has shown that microbubbles themselves can attenuate acoustic signals through scattering and absorption of the acoustic energy during both transmit and receive [51]. Given the nonlinearity of MB attenuation and the complexity of bubble-bubble interactions, further work is required to determine the exact mechanism for the peak acoustic emission observed here. The maximal concentration corresponds to the optimum between these two opposing trends. Interestingly, these optimal concentrations roughly correspond to the physiologically relevant dose for commercial ultrasound contrast agent MBs [52].

### 3.6. Harmonic Cavitation Dose vs. Gas Volume Fraction

In the spirit of using the MVD metric from prior pharmacological and therapeutic studies [26,27], we converted the MB number concentration to gas volume fraction (GVF), essentially matching total gas volume rather than the number of MBs. Figure 8 shows that as GVF increased, the sub/ultraharmonic response (HCD) rose to a peak and then declined. Importantly, there was a clear overlap between all size distributions.

At matching MI (Figure 8 left column), all MB size distributions had a similar onset, peak and decline. For all MB sizes, HCD peaked at a GVF ~0.02 µL/mL; this optimal GVF fell slightly above the estimated initial blood GVF at the recommended dose of clinically approved UCAs [53] (black dotted lines) and on the lower end of the current doses used for therapeutic MB + FUS [26] (pink dotted lines).

At matching MB size (Figure 8 right column), the trend between HCD and GVF is elucidated as a family of curves that increase with MI. Plotting as GVF aligns the onsets, peaks, and declines for all four size distributions. Interestingly, the peak HCD is approximately the same magnitude (0.0325–0.05 V^2^·s) for all sizes and occurs at a similar MI (0.4–1.0). The family of curves for 3 µm diameter MBs is remarkable in that a clear triangular shape (owing to the onset, liner increase, peak, and linear decline) is observed for HCD vs. GVF for all MIs; this should greatly simplify the application of feedback control strategies.

### 3.7. Broadband Cavitation Dose vs. Gas Volume Fraction

Figure 9 shows the broadband response as the gas volume fraction was increased. Similarly, to the sub/ultra-harmonic response, there is a gas volume fraction where the onset occurs. The onset is consistently higher than the respective onset of sub/ultra-harmonic emissions. As gas volume fraction is increased, the broadband response increases to a peak and begins to plateau or decrease. In Figure 9A,C,E,G, all microbubble size distributions have similar onset, peak and declines relative to gas volume fraction. The magnitudes of BCD are also very similar for each MB population at matching MI and gas volume fractions.

### 3.8. Master Surface Plot of Harmonic and Broadband Caviation Dose vs. Gas Volume Fraction vs. Mechanical Index

In Figure 10, all four microbubble types (2, 3, 5 µm diameter and polydisperse) were combined into a single 3D mesh to illustrate the master curve alignment for both HCD and BCD. Given how both HCD and BCD (Figure 10) show similar reductions to a single curve, resonance comes to question; it is well known that microbubble diameter influences the resonance frequency of the MB; although we do not see significant differences between the larger 5 µm MBs and smaller 2 µm MBs, even though at 1.515 MHz driving frequency we would expect the larger MBs to be closer to resonance [2,54,55,56,57]. The resonance effect could be lost due to the dynamic behavior of the MBs from pulse to pulse; it has been determined that the resonance curve and subsequent peak correspond to how much pressure is applied to the microbubble [20]. During the pulse, the MBs may fragment and/or dissolve [3,58,59], coalesce [60], displace by primary radiation force [61,62,63] and/or aggregate by secondary radiation force [63,64,65]. Thomas et al. demonstrated as MBs decayed due to ultrasound the MBs would move through their resonance peak [66]. The amount of substance that each shell lost was dependent on how high of pressure the microbubbles were being driven at [66]; these phenomena are not mutually exclusive. Additionally, depending on the driving frequency and peak negative pressure, MB oscillations can extend beyond the ultrasound pulse [58]. The change in MB number and/or diameter during the subsequent pulses may therefore change.

This study had a few limitations. First, we only investigated a single microbubble composition. As discussed previously, MB composition can greatly influence the acoustic response and colloidal stability. Although we have significant differences in size, we only apply a single frequency, 1.515 MHz; this prevents the illustration of how moving towards or away from resonance would influence the cavitation echoes for given size distribution. Similarly, our size isolation process was completed using differential centrifugation, which can result in wider size distributions than other methods such as microfluidic flow focusing. Although we observed some overlap between the intermediate 3 µm diameter MB size distribution and the others, the 2 and 5 µm distributions were not overlapping. However, when matched to the same GVF, these latter two distributions gave similar results. Thus, we do not expect that more monodisperse distributions will yield a different conclusion. However, this should be tested experimentally in future work.

## 4. Conclusions

In this article, we discovered that all four microbubble size distributions (2, 3, 5 µm diameter and polydisperse) align to similar onset and peak in both harmonic and broadband cavitation doses when matched for microbubble gas volume fraction, rather than number concentrations. Thus, microbubble gas volume fraction, a precursor of the microbubble volume dose (MVD), appears to be sufficient to capture variations in the microbubble size distribution. MVD may therefore be used as a single microbubble input, along with the derated mechanical index to represent the ultrasound input, to predict the resulting bubble activity as determined by passive cavitation detection; this result provides a missing link between prior literature showing that pharmacokinetics and blood-brain barrier disruption also scale with MVD. Future work should determine if this effect holds in vivo, and if other parameters held constant in this study (e.g., bubble chemistry and ultrasound pulsing scheme) affect the harmonic and broadband cavitation doses.

## Figures and Tables

**Figure 1 pharmaceutics-14-01925-f001:**
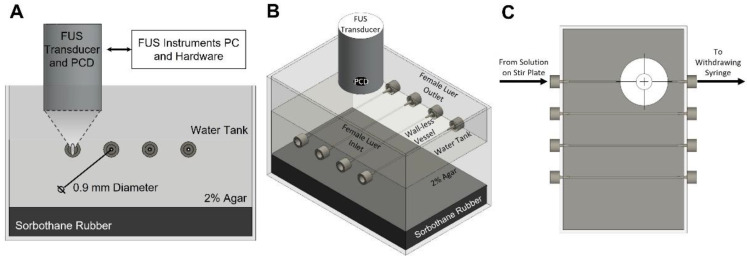
Diagram of experimental set up. A passive cavitation detector (PCD) was positioned perpendicular to the wall-less vessel and coaxially to the focused ultrasound (FUS) beam. The body of the phantom was made of 2% agar. The space between the agar and FUS transducer was degassed water. Sorbothane rubber was used beneath the agar to minimize reflections. A block diagram and side view of the phantom (**A**) shows a connection to the FUS Instruments RK50 system. An isometric view of the phantom (**B**) illustrates the four wall-less vessels with luer inlets and outlets. The microbubble flow path from solution on the stir plate through the phantom to a withdrawing syringe can be seen in the top view (**C**).

**Figure 2 pharmaceutics-14-01925-f002:**
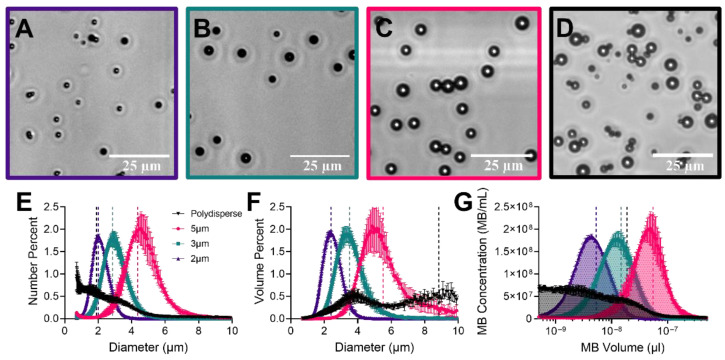
Characterization of size-selected and polydisperse microbubbles. Representative brightfield images of each diameter (2, 3 and 5 µm) and polydisperse microbubbles ((**A**–**D**), respectively). Number weighted (**E**) and volume-weighted (**F**) size distributions. (**G**) Microbubble concentration against the microbubble volume; the shaded area under the curve represents the gas volume fraction at the given total concentration (10^10^ MBs/mL). Vertical dotted lines of matched color represent mean values for the size distribution. Statistical analysis showed a significant difference (*p* < 0.05) between all isolated diameters (2, 3 and 5 µm) but not the polydisperse population. Data points represent the mean and error bars show the standard deviation (*n* = 4).

**Figure 3 pharmaceutics-14-01925-f003:**
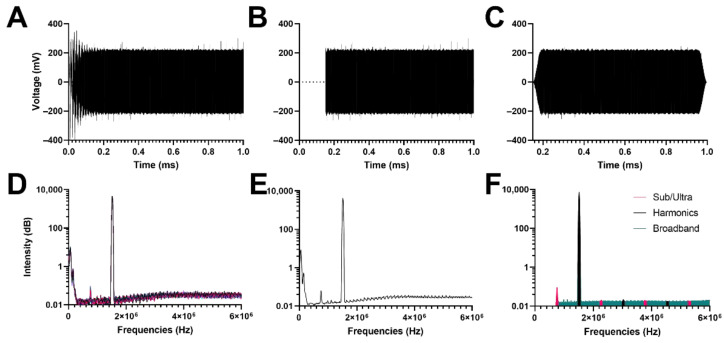
Signal processing of voltage versus time data obtained from the passive cavitation detector for a PBS control (**A**). Initial 150 µs removed to decrease noise prior to first sonication reflection at the vessel (**B**). Tukey window is applied (**C**). The Fast Fourier Transform (FFT) was taken for all 26 sonications (**D**). For all replicates, the average FFT was obtained (**E**). The area under the curve for sub/ultra-harmonics, harmonics, and broadband peaks were calculated (**F**).

**Figure 4 pharmaceutics-14-01925-f004:**
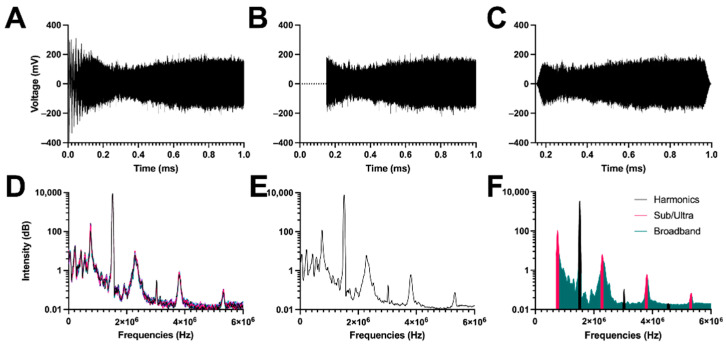
Signal processing of voltage versus time data obtained from the passive cavitation detector for 1.5 × 10^6^ MBs/mL of 5 µm MBs sonicated at 1.0 mechanical index. (**A**–**F**) Similar signal processing as Figure 3.

**Figure 5 pharmaceutics-14-01925-f005:**
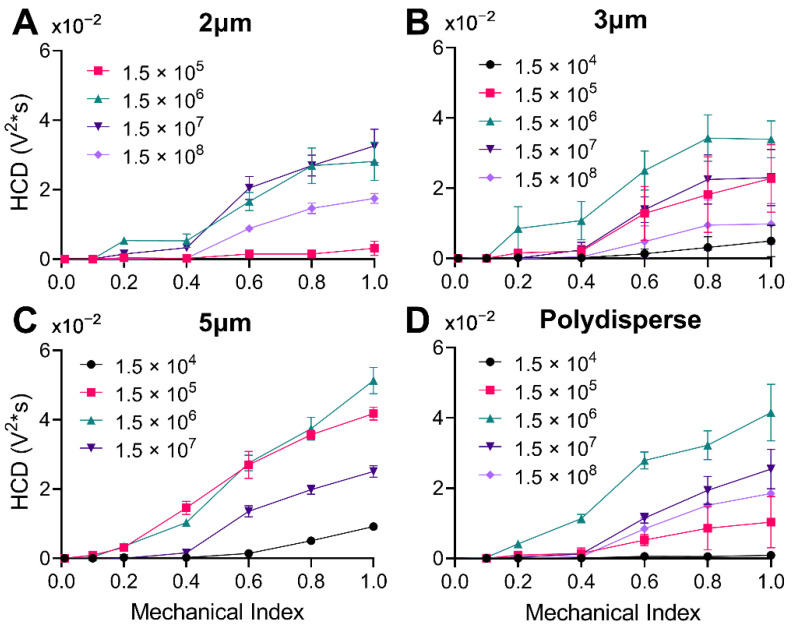
Harmonic Cavitation Dose (HCD) versus mechanical index at varying concentrations. (**A**) 2 µm microbubbles at four concentrations (1.5 × 10^5^, 1.5 × 10^6^, 1.5 × 10^7^ and 1.5 × 10^8^ MBs/mL). (**B**) 3 µm microbubbles at five concentrations (1.5 × 10^4^–1.5 × 10^8^ MBs/mL). (**C**) 5 µm microbubbles at four concentrations (1.5 × 10^4^–1.5 × 10^7^ MBs/mL). (**D**) Polydisperse microbubbles at five different concentrations (1.5 × 10^4^–1.5 × 10^8^ MBs/mL). Statistical significance can be found in Appendix A. Data represent the mean and errors bars show the standard deviation (*n* = 4).

**Figure 6 pharmaceutics-14-01925-f006:**
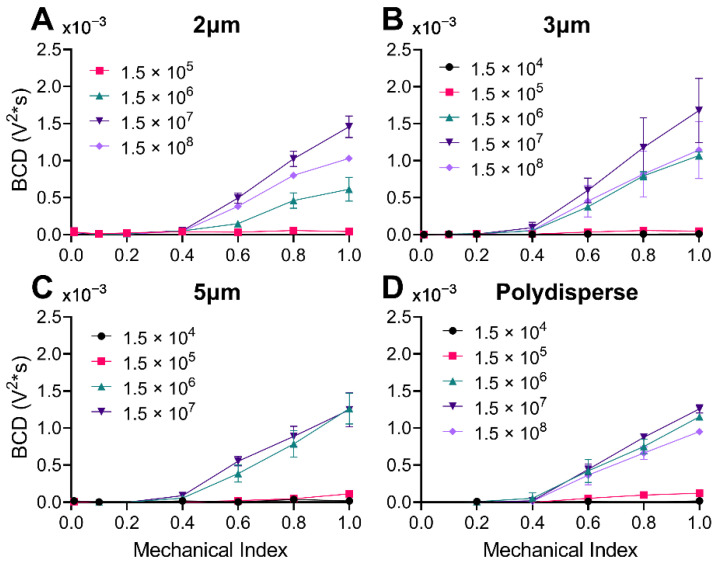
Broadband Cavitation Dose (BCD) versus the mechanical index. (**A**) 2 µm microbubbles at four concentrations (1.5 × 10^5^–1.5 × 10^8^ MBs/mL). (**B**) 3 µm microbubbles at five concentrations (1.5 × 10^4^–1.5 × 10^8^ MBs/mL). (**C**) 5 µm microbubbles at four concentrations (1.5 × 10^4^–1.5 × 10^7^ MBs/mL). (**D**) Polydisperse microbubbles at five different concentrations (1.5 × 10^4^–1.5 × 10^8^ MBs/mL). Statistical analyses can be found in Appendix A. Data represent the mean and error bars show the standard deviation (*n* = 4).

**Figure 7 pharmaceutics-14-01925-f007:**
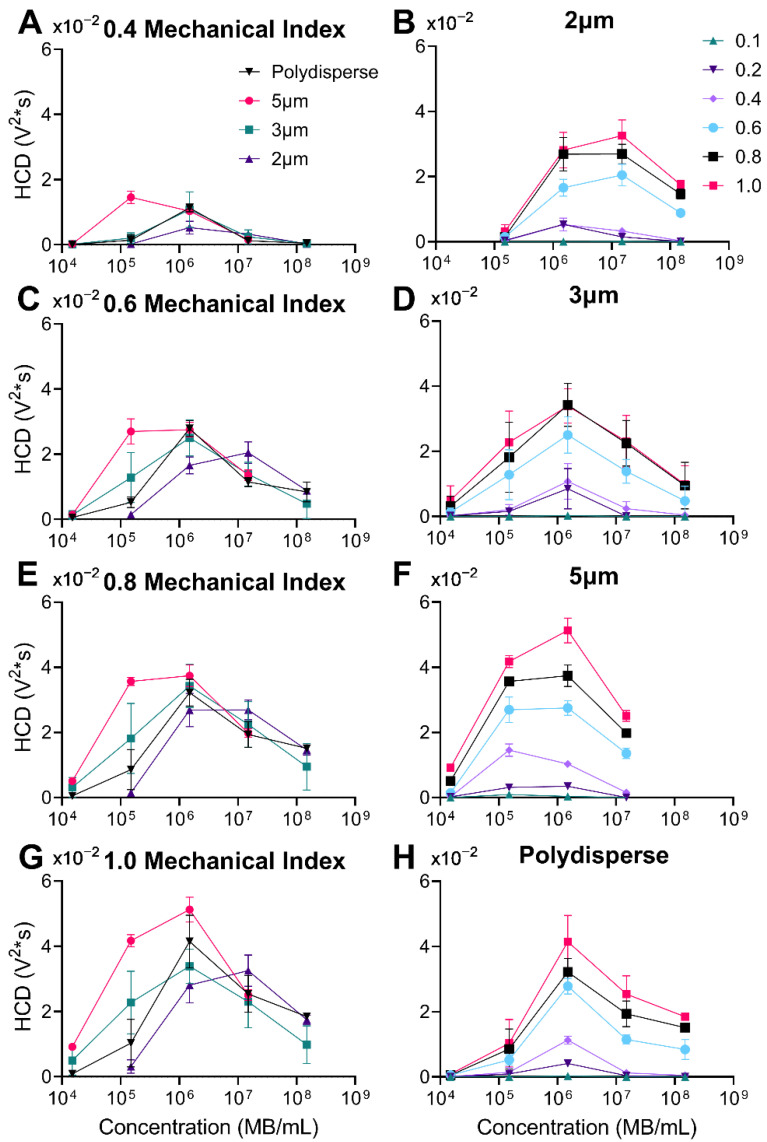
Harmonic Cavitation Dose (HCD) versus microbubble number concentration (MBs/mL). (**A**,**C**,**E**,**G**) Graphs represent the four types of microbubbles at different mechanical indices: 0.4, 0.6, 0.8, and 1.0, respectively. The inverse is shown in graphs (**B**,**D**,**F**,**H**) where each graph represents the range of mechanical indices (0.1–1.0) for each microbubble population: 2, 3, 5 µm, and polydisperse, respectively. Statistical significance can be found in Appendix A. Data represent the mean and errors bars show the standard deviation (*n* = 4).

**Figure 8 pharmaceutics-14-01925-f008:**
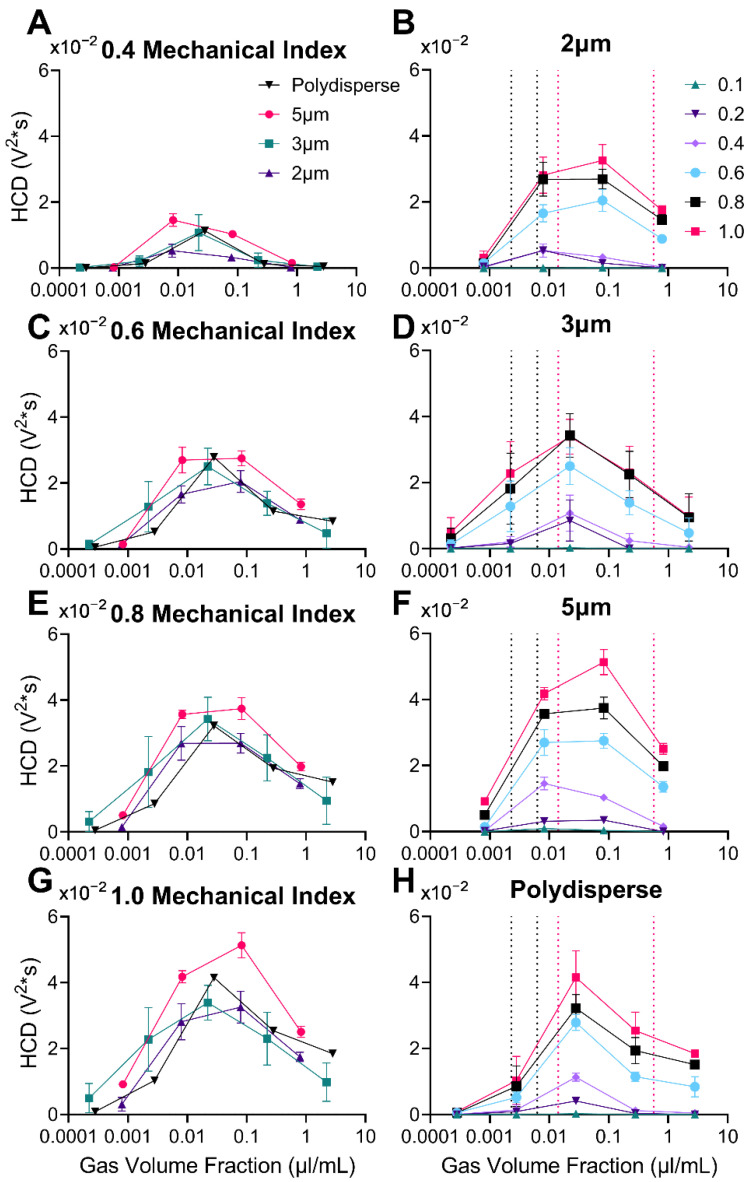
Harmonic Cavitation Dose (HCD) versus gas volume fraction (µL/mL). (**A**,**C**,**E**,**G**) Graphs represent the four types of microbubbles at different mechanical indices: 0.4, 0.6, 0.8, and 1.0, respectively. The inverse is shown in graphs (**B**,**D**,**F**,**H**) where each graph represents the range of mechanical indices (0.1–1.0) for each microbubble population: 2, 3, 5 µm diameter, and polydisperse respectively. Vertical dotted lines represent regions of gas volume fraction used in ultrasound imaging [53] (black) and blood–brain barrier disruption [26] (pink). Data represent the mean and error bars show the standard deviation (*n* = 4).

**Figure 9 pharmaceutics-14-01925-f009:**
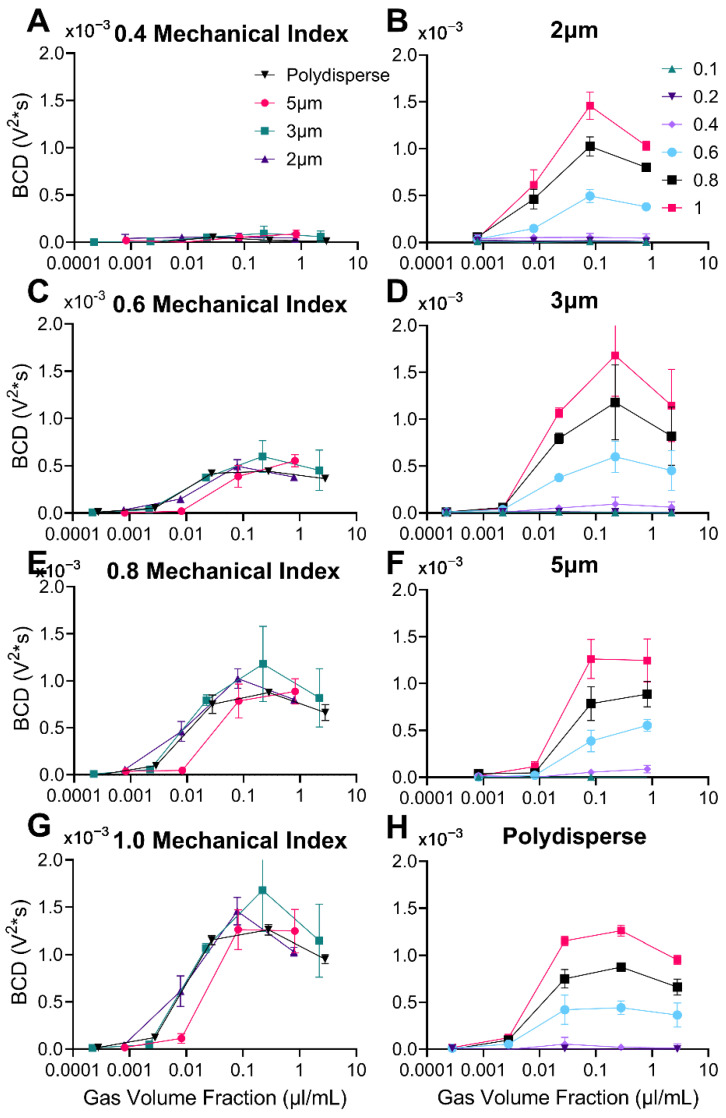
Broadband Cavitation Dose (BCD) versus gas volume fraction (µL/mL). (**A**,**C**,**E**,**G**) Graphs represent the four types of microbubbles at different mechanical indices: 0.4, 0.6, 0.8, and 1.0, respectively. The inverse is shown in graphs (**B**,**D**,**F**,**H**) where each graph represents the range of mechanical indices (0.1–1.0) for each microbubble type: 2, 3, 5 µm diameter, and polydisperse, respectively. Statistical significance can be found in Appendix A. Data represent the mean and error bars show the standard deviation (*n* = 4).

**Figure 10 pharmaceutics-14-01925-f010:**
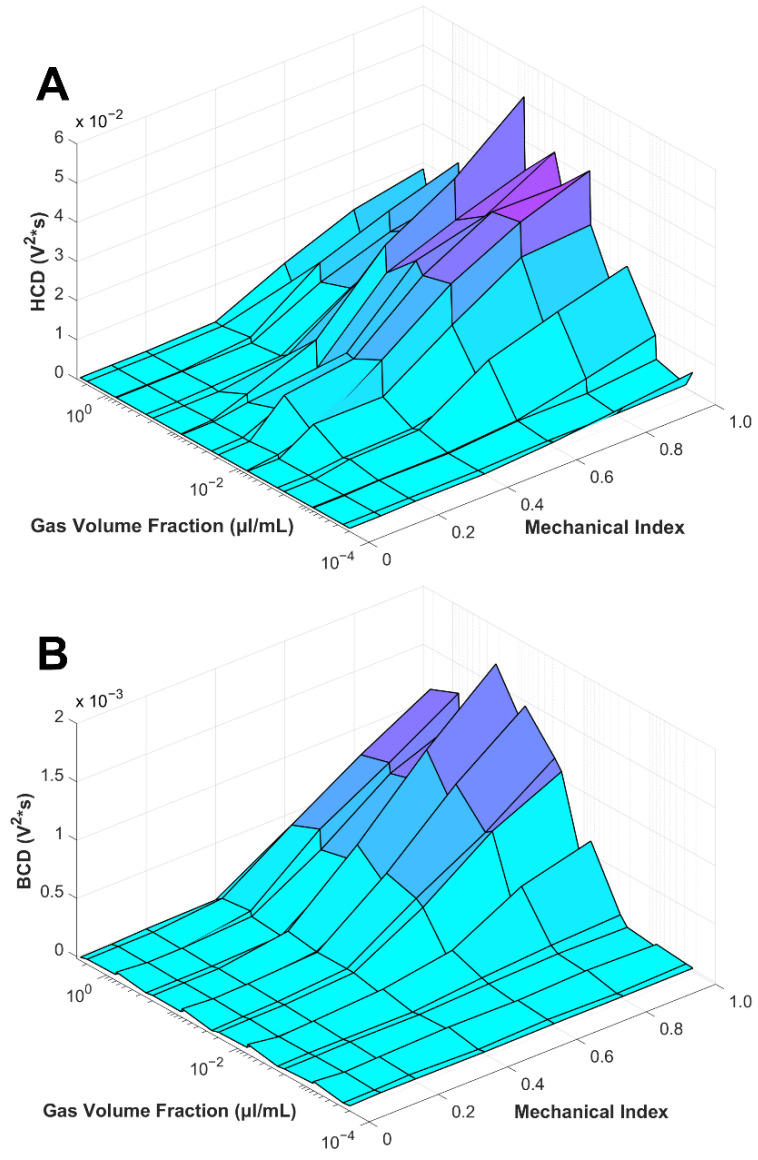
Harmonic (**A**) and Broadband (**B**) Cavitation Dose (HCD and BCD) versus mechanical index versus gas volume fraction (µL/mL). (**A**,**B**) Graphs represent all four size distributions (2, 3, 5 µm diameter and polydisperse). Error bars are not shown but can be found in previous Figure 5, Figure 6, Figure 8 and Figure 9). Statistical significance can be found in Appendix A.

**Table 1 pharmaceutics-14-01925-t001:** Microbubble size distribution statistics. Table shows the mean diameter, mode diameter, standard deviation, d_10_ and d_90_ (where 10 or 90 percent of all particles fall below this diameter), in the number weighted graph (Figure 2E). The mean diameter for the volume weighted plot (Figure 2F) is shown next. The average gas volume fraction (
ΦMB
) per microbubble is given in the rightmost column and found using Figure 2G. All values were averaged over 4 separate measurements.

Size Distribution Characterization
	Number % Parameters	Volume % Mean	*ΦMB* @ 10^10^ MBs/mL
Mean	Mode	SD	d_10_	d_90_
**2 µm**	2.0 µm	1.8 µm	0.46 µm	1.3 µm	2.4 µm	2.4 µm	5.28 µL/mL
**3 µm**	2.9 µm	2.9 µm	0.7 µm	2.2 µm	3.6 µm	3.5 µm	14.8 µL/mL
**5 µm**	4.4 µm	4.2 µm	1.1 µm	3.2 µm	5.4 µm	5.5 µm	54.7 µL/mL
**Polydisperse**	1.9 µm	1.0 µm	1.6 µm	0.8 µm	3.6 µm	8.8 µm	18.7 µL/mL

## Data Availability

The data presented in this study are available on request from the corresponding author. The data are not publicly available due to the large size of raw data collected.

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
