# Peer review of "Cavitation Characterization of Size-Isolated Microbubbles in a Vessel Phantom Using Focused Ultrasound"

_pharmaceutics, 2022, doi:10.3390/pharmaceutics14091925_

Round 1

Reviewer 1 Report

The subject investigated by the authors is very interesting. The future of the bubble based medical applications of ultrasound maybe size isolated or monodisperse stable microbubbles. Although currently only a few volatile agents are approved for clinical use. Currently there is no comprehensive knowledge on the size dependent performance of MBs in therapy applications.  The study looks at the influence of size and concentration on the pressure dependent scattering of size isolated MBs and there are interesting and important conclusions. The exposure parameters are within many  of the current therapy applications. I find the paper timely and interesting; however, there are points that need to be addressed before publication.

1-      MBs withdrawal using a syringe from the stirring beaker helps in reducing the size inhomogeneity due to different MBs rise speed. However, inside the vessel, the bigger MBs raise fast to the top of the vessel. Thus, comparisons may be difficult as there will be higher concentration of the bigger MBs at the top of the vessel wall compared to the smaller MBs cases. This may create shielding given the relatively large diameter of the vessels that is used. Please comment on how you tackled this problem. Also, please comment on the flow speed and the rationale behind it.

21-      At low concentrations, MBs inside the beaker may dissolve fast to smaller MBs. Please comment on the size stability of the MBs during the experiments. How long did every measurements last?

3-      The mean size of the MBs were different (2,3 and 5um). However, the distributions were relatively broad and there were some size distribution overlap. For example in Fig. 2E, the 2um and the 3um bubbles some overlap. Please comment on if the observed effect could be different in case of more monodisperse sizes such as those in REF[1-4] (please note that exposure parameters in these studies are more imaging based).

42-      The literature on the bubble-bubble interaction, which becomes significantly important at higher concentrations, is not complete. There are numerical studies that addressed similar phenomenon observed in more realistic scenarios of randomly distributed bubbles in space REF[5-7]. As an instance, in your paper you state: “The simulations showed that bubble-bubble interactions intensified broadband signaling and therefore hid the harmonic emissions [43,44]. Additionally, previous work has shown that microbubbles themselves can attenuate acoustic signals through scattering and absorption of the acoustic energy during both transmit and receive [45].”- Citing these studies were relevant for the bubble-bubble interactions. However, the simulation studies used an approximate form for the bubble-bubble interaction and they do not consider a more realistic case of randomly distributed bubbles in space. Similar observations on the signal strength are made in REF[5-7] in the regime of stable oscillations. Study in REF[4] specifically considered the case of the encapsulated MBs and in clinically relevant MB doses  and showed that increasing concentration leads to plateauing and possible decrease of the  SH and UH strength. Similar to the finding of the current manuscript they reported that “This indicates that there exists an optimum concentration for the maximum SH and UH emissions.” Ref[7] reported the saturation and decrease of the scattered signal with increasing concentration. In fact, the value of ~1e6MBs/ml predicted for MBs as the optimum dose is consistent with the experimental study here as well as the reported increased optimum concentration with decreasing MB size. Citing these works and similar other relevant works complements the importance of the findings related to concentration. In fact, I would highlight the concentration related observation in this paper. The finding can be relevant to many pre-clinical studies that uses very large MB doses.

53-      Varying the bubble concentration in this work and the associated findings were important. However, some of the concentrations (e.g. 1.5E8MBs/ml) may seem much higher than the clinically approved doses. Please comment on this and provide some comparisons with the approved doses for better understanding of the general reader. At the end you nicely concluded that the optimum dose correlates well with the clinically relevant dose in [40].  However, this is not analyzed quantitavely.

64-      Please be careful of relating the decrease in the scattered signal to increased attenuation. While, increased attenuation reduces the pressure that reaches the bottom layer bubbles; however, the relationship between pressure and attenuation is nonlinear. Thus, as the wave propagates towards the bottom layer, due to the pressure change the attenuation may also drop. Moreover, bubble-bubble interaction reduces the attenuation (in a similar fashion to scattering), so attenuation doesn’t linearly increase with increasing concentration. The reduction in signal is a factor of both the suppression of bubble oscillations due to increased interaction and possible attenuation increase.

75-      There are some recent studies that show the advantage of size selected MBs, although in the imaging settings REF[1-4]. Possible discussion of these studies will help readers better understand the importance of the size isolation approach in the paper.

86-      The limitations of this study are not reported. They should be clearly enumerated.

77-      In conclusion “align to a similar onset and peak in both harmonic and broad-436 band cavitation doses when matched for microbubble gas volume fraction.” Can you comment on the possible physical mechanism? Can this be die to the fact that you sonicated these bubbles below their resonance frequency and thus, Blake threshold (which is very similar for the considered sizes between 105-120kPa) was more of a  determining factor for the onset? As such, the onsets were similar? In REF[1-4] bubbles are sonicated by their resonance or pressure dependent resonances thus the onset effects were more obvious.  

 8-      “ We therefore use the terms “harmonic” or “broad-band” cavitation, rather than “stable” or “inertial” cavitation, to describe the behavior”-  I completely agree with the authors on the reasoning behind this. Please highlight this better in the manuscript. Intermittent inertial cavitation is associated with persistent sub and ultra harmonic peaks, however it cannot be considered stable.

29- Please put the flow and the vessel diameter details in fig. 1. Also mention the reason behind using the specific 900um diameter vessel and the 34ml/hr. For the vessel diameter used in this paper the arterial blood flow rate is more than 3ml/min and more than 1.2 ml/min in veins REf[9]

       10- Intensity-frequency curves are better to be illustrated in dB scale. This will help for better illustration of the sub/ultra and harmonic peaks

111-Figure 3 caption is not on the same page as of the manuscript. Please fix it and everywhere else in the manuscript.

112-"Additionally, at a given MI, the HCD increased to a peak and then decreased with MB concentration, presumably due to attenuation of the US signal during both transmit and receive.”- Attenuation increased not accurate.. Please mention bubble-bubble interaction as well. It is shown.

113- Concentrations are reported as MB/mL. This can change to MBs/mL through out the whole manuscript. Also unit of concentration is missed in figure captions.

1   14- Line 385-simply should be simplify

    15- There are studies that reported the influence of the size and shell on the performance of MBs in medical ultrasound (e.g ref 16 in the paper and Ref[8] here). How does your findings relate to the in vivo studies?

 References

1-  Segers, T., De Jong, N. and Versluis, M., 2016. Uniform scattering and attenuation of acoustically sorted ultrasound contrast agents: Modeling and experiments. The Journal of the Acoustical Society of America, 140(4), pp.2506-2517.

2- Segers, T., Kruizinga, P., Kok, M.P., Lajoinie, G., De Jong, N. and Versluis, M., 2018. Monodisperse versus polydisperse ultrasound contrast agents: Non-linear response, sensitivity, and deep tissue imaging potential. Ultrasound in medicine & biology, 44(7), pp.1482-1492.

3- Jafari Sojahrood, A., de Leon, A.C., Lee, R., Cooley, M., Abenojar, E.C., Kolios, M.C. and Exner, A.A., 2021. Toward precisely controllable acoustic response of shell-stabilized nanobubbles: High yield and narrow dispersity. ACS nano, 15(3), pp.4901-4915.

4- Helbert, A., Gaud, E., Segers, T., Botteron, C., Frinking, P. and Jeannot, V., 2020. Monodisperse versus polydisperse ultrasound contrast agents: In vivo sensitivity and safety in rat and pig. Ultrasound in Medicine & Biology, 46(12), pp.3339-3352.

5- Sojahrood, A.J., Earl, R., Haghi, H., Li, Q., Porter, T.M., Kolios, M.C. and Karshafian, R., 2021. Nonlinear dynamics of acoustic bubbles excited by their pressure-dependent subharmonic resonance frequency: influence of the pressure amplitude, frequency, encapsulation and multiple bubble interactions on oversaturation and enhancement of the subharmonic signal. Nonlinear Dynamics, 103(1), pp.429-466.

6- Haghi, H., Sojahrood, A.J. and Kolios, M.C., 2019. Collective nonlinear behavior of interacting polydisperse microbubble clusters. Ultrasonics sonochemistry, 58, p.104708.

7- Haghi, H., Jafari Sojahrood, A., De Leon, A.C., Exner, A. and Kolios, M.C., 2018. Experimental and numerical investigation of backscattered signal strength from different concentrations of nanobubble and microbubble clusters. The Journal of the Acoustical Society of America, 144(3), pp.1888-1888.

8- McMahon, D., Lassus, A., Gaud, E., Jeannot, V. and Hynynen, K., 2020. Microbubble formulation influences inflammatory response to focused ultrasound exposure in the brain. Scientific reports, 10(1), pp.1-15.

9- Klarhöfer, M., Csapo, B., Balassy, C., Szeles, J.C. and Moser, E., 2001. High‐resolution blood flow velocity measurements in the human finger. Magnetic Resonance in Medicine: An Official Journal of the International Society for Magnetic Resonance in Medicine, 45(4), pp.716-719.

Reviewer 2 Report

The authors here did a comprehensive study of microbubble size versus passive cavitation dose signal. The study fits the special issue well and is important to the readers interested in ultrasound-mediated drug delivery. A few minor points can be improved to increase the reader experiences and enhance the reproducibility of the study.

1. Method, Line 164: the authors should provide more detail about the method of separating polydisperse MB and generating MBs with different size ranges (e.g. specify what centrifuge and speed was used) as these are the major parameters in this manuscript. Also, the authors should specify how the MBs were diluted for the concentration experiments.

2. Result, Page 7: The authors reported they had isolated varied sizes of MBs for the follow-up study. Therefore, it would be better if the authors can report not just the mean value but also the SD, mode, or even 95% size range of the MB prep.

3. Supplementary Table: The authors should indicate what the colour code on the table means.

Round 2

Reviewer 1 Report

The authors addressed all my comments and the work can be published following a minor revision.

Please fix the quality of figure S2. It is not readable.

Author Response

Figure 2 has been re-inserted to make it readable.